# Rev-erbα Inhibits Proliferation and Promotes Apoptosis of Preadipocytes through the Agonist GSK4112

**DOI:** 10.3390/ijms20184524

**Published:** 2019-09-12

**Authors:** Guiyan Chu, Xiaoge Zhou, Yamei Hu, Shengjie Shi, Gongshe Yang

**Affiliations:** 1Key Laboratory of Animal Genetics, Breeding and Reproduction of Shaanxi Province, Yangling 712100, China; 2Laboratory of Animal Fat Deposition & Muscle Development, College of Animal Science and Technology, Northwest A&F University, Yangling 712100, China

**Keywords:** Rev-erbα, GSK4112, proliferation, apoptosis, β-catenin

## Abstract

Proliferation and apoptosis are important physiological processes of preadipocytes. Rev-erbα is a circadian clock gene, and its activity contributes to several physiological processes in various cells. Previous studies demonstrated that Rev-erbα promotes preadipocyte differentiation, but a role of Rev-erbα on preadipocyte proliferation and apoptosis has not been demonstrated. GSK4112 is often used as an agonist of Rev-erbα. In this study, we used GSK4112 to explore the effects of Rev-erbα on preadipocyte proliferation and apoptosis by RT-qPCR, Western blot, Cell Counting Kit-8 (CCK8) measurement, 5-Ethynyl-2′-deoxyuridine (EdU) staining, Annexin V-FITC/PI staining, and flow cytometry. These results revealed that GSK4112 inhibited the viability of 3T3-L1 preadipocytes and decreased cell numbers. There was also decreased expression of the proliferation-related gene Cyclin D and the canonical Wingless-type (Wnt) signaling effect factor β-catenin. Furthermore, palmitate (PA)-inducing cell apoptosis was promoted. Overall, these results reveal that Rev-erbα plays a role in proliferation and palmitate (PA)-inducing apoptosis of 3T3-L1 preadipocytes, and thus may be a new molecular target in efforts to prevent and treat obesity and related disease.

## 1. Introduction

Obesity is a global public health problem and often occurs with type 2 diabetes and nonalcoholic fatty liver disease (NAFLD) [1,2]. Obesity is primarily caused by excessive expansion of white adipose tissue, with increased size and number of adipocytes [3,4]. Due to the increasing obesity problem, research and development of effective measures to control and cure obesity are becoming increasingly urgent [5]. Overall, it is important to understand the molecular mechanisms of obesity for effective prevention and treatment of obesity and related metabolic diseases [6].

Rev-erbα is a nuclear receptor that can act as a transcriptional repressor [7,8]. In mammals, Rev-erbα is highly expressed in various organ tissues, including liver [9], skeletal muscle [10], brain [11], and adipose tissue [12], where it participates in development and circadian regulation. Rev-erbα can function as a nuclear receptor to convert external nutrient signals into specific gene regulation [13,14]. Rev-erbα regulates many biological processes, including cell differentiation [15], proliferation [16], apoptosis and autophagy [10,17], metabolic balance of carbohydrates and lipids [18], bile acid synthesis [19], and cholesterol homeostasis [20]. Mice lacking Rev-erbα showed increased adiposity. After consuming a high-fat diet (HFD), Rev-erbα^−/−^ mice were overweight and exhibited hyperleptinemia, hyperlipidemia, hypercholesterolemia, hyperinsulinemia, and fatty liver [21]. In 3T3-L1 cells, Rev-erbα promoted adipocyte differentiation [22] and 3T3-L1 preadipocyte differentiation using both natural and synthetic ligands [23,24,25]. Rev-erbα also acts in cell proliferation. Study of the role of Rev-erbα in cancer cells has shown that the pharmacological activation of Rev-erbα by its agonists SR9009 and SR9011 was lethal to various cancer cells [17]. Rev-erbα induced apoptosis in cancer cells by regulating autophagy and inhibiting de novo lipogenesis, causing death of cancer cells. These findings confirmed the role of Rev-erbα in cancer cell proliferation but the role of this protein in 3T3-L1 preadipocyte proliferation and apoptosis still needs to be explored.

Wingless-type (Wnt)/β-catenin signaling plays an important role in cell proliferation [26,27]. Activation of the Wnt signaling pathway promotes the accumulation of canonical factor β-catenin in the cytoplasm. β-catenin translocates into the nucleus to activate transcription factors in the T-cell factor/lymphoid enhancer factor (TCF/LEF) family [28], which then stimulate expression of downstream genes such as Cyclin D1 that play important roles in regulating cell cycle progression [29]. A previous study found that the inhibition by Rev-erbα of bone mesenchymal stem cell (BMSC) proliferation was partially reversed when Wnt/β-catenin signaling was activated, suggesting that Rev-erbα might regulate cell proliferation through Wnt/β-catenin signaling [16]. However, if there is a contribution of Rev-erbα to the proliferation of 3T3-L1 preadipocytes remains unclear. Investigation of Rev-erbα synthetic agonists has provided insights into Rev-erbα physiological function and also opens up the possibility of drug treatments for various diseases, including obesity. GSK4112 is a well-known agonist of Rev-erbα and acts by increasing the binding of Nuclear receptor co-repressor (NCOR) to Rev-erbα, which inhibits transcription of Rev-erbα target genes [7,30]. This study explored the effect of Rev-erbα agonist GSK4112on the proliferation of 3T3-L1 preadipocytes.

Our results showed that GSK4112 restrained proliferation of 3T3-L1 preadipocytes and β-catenin expression was decreased. As predicted, palmitic acid (PA)-induced apoptosis was enhanced by GSK4112. These results implied that Rev-erbα is a reasonable molecular target for treatments aiming to prevent or treat obesity and related diseases.

## 2. Results

### 2.1. The Rev-erbα Agonist GSK4112 Inhibited Pre-Adipocyte Viability

Rev-erbα is a nuclear receptor whose function requires its corresponding ligand, heme. Kumar et al. treated 3T3-L1 cells in the differentiation stage and observed effects with 10 µM of the Rev-erbα synthetic ligand, GSK4112 [25]. Therefore, we selected this concentration of GSK4112 to ask if Rev-erbα affects proliferation and apoptosis of 3T3-L1 cells. The results in Figure 1A show effective action of GSK4112, as evidenced by decreased Bmal1 expression [31]. Further detection of cell viability by measuring CKK-8 levels revealed significant suppression of cell viability by GSK4112 treatment for 24 h and 48 h compared with the DMSO treatment group (Figure 1B).

### 2.2. The Rev-erbα Agonist GSK4112 Inhibited Cell Proliferation

In order to determine whether Rev-erbα affects the proliferation process of 3T3-L1 cells, we treated 3T3-L1 cells with GSK4112 for 24 h. The results of 5-Ethynyl-2′-deoxyuridine (EdU) staining showed that GSK4112 application decreased the percentage of positive cells (red/green) compared with the DMSO group (Figure 2A,B). The cell cycle distribution was measured by flow cytometry and the results indicated that GSK4112 effectively inhibited the transition from G1-Phase to S-phase (Figure 2C,D). Thus, GSK4112 inhibited cell proliferation and decreased cell number.

### 2.3. Rev-erbα Inhibited Proliferation of 3T3-L1 Cells through the Wnt Signaling Pathway

To explore how Rev-erbα affects the proliferation of 3T3-L1 cells, we next measured the expression of related genes. As expected, we found that GSK4112 obviously suppressed the expression of the proliferation-promoting factor Cyclin D at both the RNA and protein levels (Figure 3A,B). Additionally, GSK4112 promoted expression of an inhibitor of proliferation, p27 (Figure 3B). GSK4112 also inhibited expression of the canonical Wnt signaling pathway effect factor β-catenin (Figure 3C,D). These results suggested that Rev-erbα might affect the 3T3-L1 cell proliferation process by interaction with the Wnt signaling pathway.

### 2.4. Cell Model of Palmitate-Induced 3T3-L1 Preadipocyte Apoptosis

When proliferation is blocked, cells may initiate the apoptosis program [32]. In order to further explore whether GSK4112 not only blocks the proliferation of cells, but also promotes apoptosis, we next measured cell apoptosis through cell staining and measurement of apoptosis-related gene expression. To do this, cells were incubated with 250 µM PA for 8 h, 12 h, or 24 h. PA treatment for 24 h increased the mRNA levels of Bax and Caspase-3, but suppressed the level of Bcl-2 (*p* < 0.01) (Figure 4C). These data demonstrated a successful cell model of palmitate-induced 3T3-L1 preadipocyte apoptosis. Interestingly, PA also elevated the mRNA level of Rev-erbα (Figure 4D).

### 2.5. Rev-erbα Agonist GSK4112 Aggravated Palmitate-Induced Preadipocyte Apoptosis

In order to detect whether Rev-erbα induces apoptosis, GSK4112 was used to stimulate Rev-erbα activity after PA treatment. To do this, 3T3-L1 cells were first incubated with 0.25 mM PA for 12 h, then 10 µM GSK4112 was added for 24 h. Annexin V/PI staining and flow cytometry analysis revealed a lower percentage of live cells and a greater number of cells in the early apoptosis stage after GSK4112 treatment as compared with the DMSO treatment group (Figure 5A,B). In addition, GSK4112 treatment increased the mRNA levels of Bax and Caspase-3 and reduced the level of Bcl-2 (Figure 5). At the protein level, GSK4112 increased the relative ratio of Bax/Bcl-2 together with cleaved-caspase3/caspase3, indicating successful aggravation by Rev-erbα of apoptosis in 3T3-L1 cells. The above data demonstrated that Rev-erbα promoted PA-induced apoptosis. Apoptosis-related genes like Bax and Caspase-3 showed increased expression after GSK4112 stimulation. Therefore, we speculated that Rev-erbα might promote PA-induced apoptosis in 3T3-L1 preadipocytes.

## 3. Discussion

The number of adipocytes can affect the volume of adipose tissue, and the number of adipocytes depends on the balance between cell proliferation and apoptosis [4]. Research of the loss of Rev-erbα function in mice indicated that Rev-erbα was related to obesity and related diseases [9,21]. Previous reports suggested that Rev-erbα inhibited the differentiation of 3T3-L1 preadipocytes, making it a good model to study obesity [15]. The discovery of specific ligands to stimulate function of this protein has accelerated study of the physiological function of Rev-erbα [23,33]. Here we detected the role of Rev-erbα in the proliferation and apoptosis of 3T3-L1 preadipocytes by treatment with GSK4112, an agonist for Rev-erbα.

GSK4112 was the first to be identified as a Rev-erbα-targeting ligand that increases the binding of Nuclear receptor co-repressor (NCOR) to Rev-erbα, resulting in inhibition of the transcription of Rev-erbα target genes [7,33]. Our results indicated that GSK4112 inhibited the proliferation of 3T3-L1 preadipocytes, consistent with the results of a previous study that showed that Rev-erbα can inhibit the proliferation of breast cancer cells [34]. Proliferation is determined by the cell cycle and related regulatory mechanisms [35]. Rev-erbα can regulate the circadian system or other cell cycle-related genes by binding to RRE elements in their promoter sequences [28,36]. Therefore, GSK4112 may have altered the cell cycle by changing the expression of circadian clock genes via changes in the activity of REV-erbα. The Bmal1 RNA level was inhibited, indicating effective GSK4112 treatment. The activation of Rev-erbα suppressed cell proliferation decreased mRNA and protein expression of Cyclin D and increased expression of p27 at the protein level. Cyclin D and p27 may be circadian clock-controlled genes or may have RRE elements in their promoter sequences. The canonical Wnt signaling pathway has been shown to play a positive role in proliferation in 3T3-L1 cells [37] and neural stem cells [27]. Previous studies demonstrated that the inhibition of bone mesenchymal stem cell proliferation induced by Rev-erbα overexpression was partially reversed by activation of the Wnt/β-catenin signaling [16]. In this study, we found that GSK4112 decreased the expression of β-catenin mRNA and protein during proliferation of 3T3-L1 cells. This result indicated that Rev-erbα inhibited cell proliferation, possibly through Wnt/β-catenin signaling. Additional studies are required to determine the specific mechanism of interaction between Rev-erbα and Wnt/β-catenin signaling. These results revealed that Rev-erbα may suppress the increase of adipose tissue volume by inhibiting the proliferation of adipocytes. When proliferation is blocked, cells may initiate an apoptosis program [32]. The apoptosis program is initiated to remove unneeded and dangerous cells of multicellular organisms [38]. Previous studies found that palmitic acid (PA) could induce mouse preadipocyte apoptosis by increasing the cysteinyl aspartate specific proteinase (caspase) cascade reaction [39]. In this study, we constructed an apoptosis model using palmitic acid. Using this model, we found that Rev-erbα enhanced the PA-induced apoptosis process through inhibition of Bcl-2 and promotion of Bax. Bcl-2 encodes a mitochondrial outer membrane protein and can regulate intracellular Ca^2+^ release to inhibit cell apoptosis [40,41]. The reduction of Bcl-2 disrupts the balance of Bcl-2 and Bax [40,42]. The increase in Bax releases cytochromes into the cytosol, activating the caspase cascade, including Caspase-3, which induces cell apoptosis [41]. Interestingly, Rev-erbα RNA expression also increased with PA-induced apoptosis. The GSK4112 treatment results suggest that REV-erbα may promote apoptosis by increasing the relative ratios of Bax/Bcl-2 and cleaved-caspase3/caspase3. Our results demonstrated that GSK4112 promoted PA-induced apoptosis by increasing the expression of pro-apoptosis factor Bax, increasing the activation of Caspase-3 and decreasing the expression of Bcl2. These data suggested that Rev-erbα might inhibit obesity by promoting adipocytes apoptosis to reduce the number cells or volume of adipose tissue. Further studies are warranted to explore the mechanisms by which Rev-erbα affects apoptosis.

In summary, we uncovered a novel function of nuclear receptor Rev-erbα in the proliferation of 3T3-L1 cells. The classical Wnt signaling pathway was affected with decreases in β-catenin mRNA and protein levels, and Rev-erbα aggravated PA-induced apoptosis. Our results support the targeting of nuclear receptor Rev-erbα to treat obesity and related diseases.

## 4. Materials and Methods

### 4.1. Materials

GSK4112 (TOCRIS, Bristol, UK) is a Rev-erbα agonist [23]. GSK4112 was dissolved in DMSO (Solaibao, Beijing, China) to 25 mM and the stock solutions were stored at −20 °C.

### 4.2. Cell Culture and Treatment

The 3T3-L1 cell line was obtained from the Stem Cell Bank, Chinese Academy of Sciences. Cells were incubated in Dulbecco Modified Eagle Medium (Gibco, CA, USA), 10% fetal bovine serum (FBS, Gibco, CA, USA), and 100 U/mL penicillin–streptomycin. The cells were incubated at 37 °C under humidified 5% CO2 and 95% air. The medium was changed every other day.

3T3-L1 cells were treated with GSK4112 (10 µM) or DMSO (10 µM) at 30–40% density. When the density reached 70–80%, the cells were harvested for subsequent experiments. In the apoptosis experiment, 3T3-L1 cells were first treated with 250 µM palmitate (Solaibao, Beijing, China) or bovine serum albumin (BSA, 0.5%, control). The cells were then added to GSK4112 or DMSO and incubated.

### 4.3. CCK-8 Assay

A Cell Counting Kit-8 (Dojindo, Japan) was used to measure 3T3-L1 cell viability. Cells were seeded in a 96-well plate at a density of 2 × 10^3^. The cells were treated with GSK4112 for 24 h or 48 h. To each well, 10 µL CCK-8 solution was added, followed by incubation for 2 h at 37 °C. Absorbance was measured at 450 nm wavelength using vector 5 (Waltham, MA, USA).

### 4.4. EdU Imaging Assay

3T3-L1 cells were seeded in 96-well plates at a concentration of 2 × 10^3^ per well. The 3T3-L1 cells were then treated with GSK4112 for 48 h and then incubated with 50 μM EDU (RiboBio, Guangzhou, China) for 2 h. Cells were washed twice with PBS, fixed with 4% paraformaldehyde for 30 min, neutralized with 2 mg/mL glycine for 5 min, and then permeabilized with 0.5% Trixon-100 for 5 min. At the end of each step, cells were washed twice with PBS for 5 min each time. According to the kit, the cells were incubated in the mixture of Reagent B, C, D, and E for 30 min. The cells were then washed three times with 0.5% Trixon-100 and then washed twice with methanol. The nuclei were stained with Hoechst for 30 min. The stained cells were observed using a Nikon TE2000 microscope (Nikon, Tokyo, Japan) and the data were analyzed using Image J.

### 4.5. Flow Cytometry

3T3-L1 cells were seeded in a 6- well culture plate at a density of 4 × 10^5^ cells per well. After 24 h, cells were treated with the agonist GSK4112 for 48 h. The cells were washed twice with PBS and trypsin/EDTA was used to harvest cells at 80% density. Cells were centrifuged at 1500 rpm for 5 min, then fixed with 70% alcohol overnight at −20 °C and stained with 50 mg/mL Hochest 3328 (Solaibao, Beijing, China) at 4 °C for 0.5 h. Finally, samples were subjected to flow cytometry (Becton Dickinson, Franklin Lakes, NJ, USA). The proliferative index shows the ratio of mitotic cells from 10,000 cells examined.

### 4.6. Apoptosis Assessment

Apoptosis of 3T3-L1 cells were measured using an Annexin V-FITC/PI apoptosis assay kit (LIANKE, Hangzhou, China) according to the manufacturer’s protocol. The cells were observed using a Nikon TE2000 microscope (Nikon, Tokyo, Japan) and the data were analyzed with Image J. The proportions of cells in distinct apoptosis stages were detected with flow cytometry.

### 4.7. RNA Isolation and Quantitative Real-Time PCR

Total RNA samples were isolated using Trizol (TakaRa, Otsu, Japan), and the final concentrations were measured by NanoDrop 2000 (Thermo, Waltham, MA, USA). The cDNA was synthesized using a reverse transcription kit (TakaRa, Otsu, Japan). We used real-time quantitative PCR for mRNA analysis, and every reaction was performed in triplicate using a SYBR Premix (Vazyme, Nanjing, China) on a StepOne Real-Time PCR Machine (ABI, Carlsbad, CA, USA). The relative level of mRNA was normalized to that of β-actin and calculated using the 2-∆∆Ct algorithm. Primer sequences used for RT-qPCR are listed in Table 1.

### 4.8. Western Blot Analysis

3T3-L1 cells were washed three times with PBS before adding RIPA (Beyotime, Shanghai, China) supplemented with protease inhibitors (Pierce, Rockford, IL, USA) at 4 °C. We scraped the lysed cells and then centrifuged (rpm) at 4 °C for 10 min. Supernatant protein concentration was determined by BCA protein assay kit (Cwbio, Beijing, China). A 1/4 volume of 5× loading buffer (Cwbio, Beijing, China) was added to an aliquot of the supernatant, and a 20 µg protein sample was separated using a 12% SDS-polyacrylamide gel. After electrophoresis, the gel was transferred to a polyvinylidene fluoride (PVDF) membrane (CST, Boston, MA, USA) using a current of 250 mA. After 2 h, the membrane was removed and placed in 5% skim milk for 2 h in room temperature. The membrane was then incubated with primary antibody at 4 °C overnight and then incubated with secondary antibody (Boster, Wuhan, China) for two hours in room temperature. Finally, the signals were detected by a gel imaging system (Bio-Rad, CA, USA) and analyzed by Image Lab (Bio-Rad, CA, USA) software. Primary antibodies were used against Cyclin E, Cyclin D1, p27, Bcl-2, Bax, and Caspase-3 (Santa Cruz, Dallas, TX, USA); against Cleaved Caspase-3 (abways, Shanghai, China), against β-actin (CWBio, Beijing, China), and against β-catenin (Boster, Wuhan, China).

### 4.9. Statistical Analysis

Experimental data were analyzed using GraphPad Prism 7.0 and all data results are shown as mean ± SEM. Significant differences between groups were analyzed using Student′s *t* test. Values were considered significant for *p* values less than 0.05 (*, *p* < 0.05; **, *p* < 0.01).

## Figures and Tables

**Figure 1 ijms-20-04524-f001:**
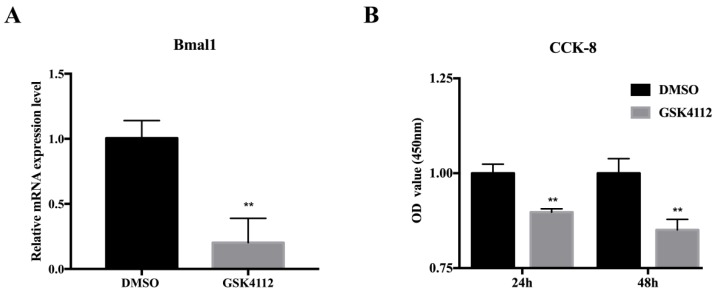
The Rev-erbα agonist GSK4112 inhibited pre-adipocyte viability. (**A**) The effect of Rev-erbα agonist GSK4112 (10 µM) on mRNA expression of Bmal1 in 3T3-L1 cells; (**B**) cell viability was detected by measuring CCK-8 levels after GSK4112 treatment for 24 h and 48 h. The absorbance values at 450 nm were measured. The statistical results represent the mean ± SEM, *n* = 3. ** *p* < 0.01.

**Figure 2 ijms-20-04524-f002:**
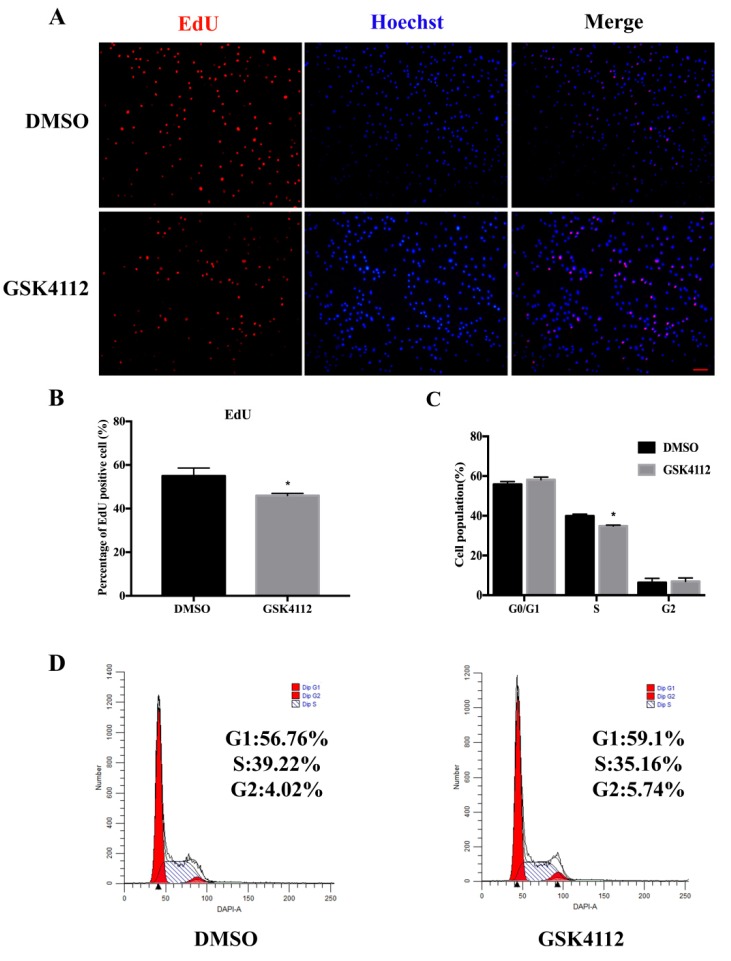
The Rev-erbα agonist GSK4112 inhibited cell proliferation. (**A**) 5-Ethynyl-2′-deoxyuridine (EdU) staining assay was carried out after GSK4112 (10 μM) treatment for 24 h. Red (EdU) stained cells indicating proliferating cell nuclei and blue (Hoechst) representing cell nuclei, scale bar 100 μm. (**B**) The results are represented as the percentage of red/blue cell nuclei. (**C**) The data statistics of Flow cytometry. (**D**) Flow cytometry was used to determine the percentages of cells in different cycle phases. The cell treatment was the same as for the EdU staining assay, and the nuclei were stained by DAPI. Statistical results are representative of the mean ± SEM of three independent experiments. * *p* < 0.05.

**Figure 3 ijms-20-04524-f003:**
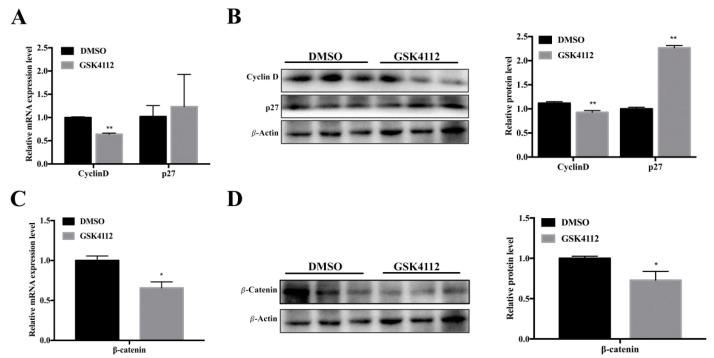
The effect of Rev-erbα agonist GSK4112 on the expression of proliferation-related genes and β-catenin. (**A**) RT-qPCR analysis of cell cycle-related genes after GSK4112 treatment for 24 h. (**B**) Western blot analysis of cell cycle-related proteins. (**C**) The mRNA expression of β-catenin was detected by RT-qPCR. (**D**) The protein expression level of β-catenin was detected by Western blot. Data are presented as mean ± SEM of three independent experiments. * *p* < 0.05; ** *p* < 0.01.

**Figure 4 ijms-20-04524-f004:**
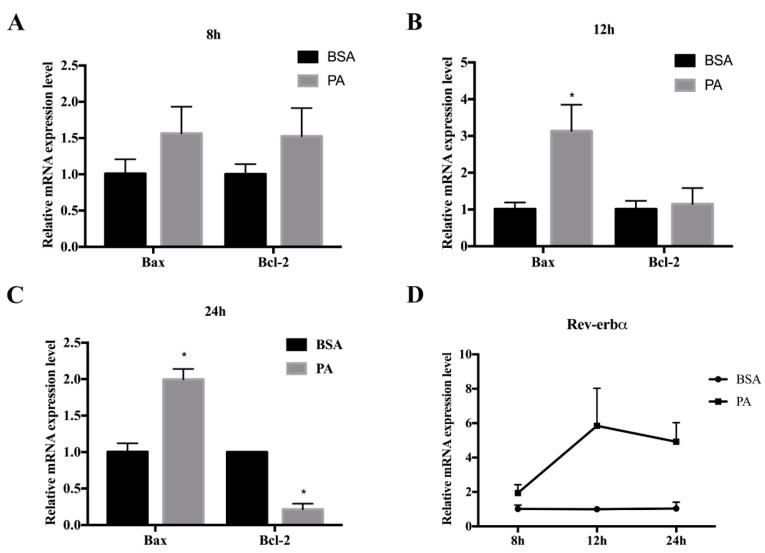
Cell model of palmitate-induced 3T3-L1 preadipocyte apoptosis. 3T3-L1 cells were induced with 250 µM palmitic acid (PA) or 0.5% BSA for 8, 12, or 24 h. The mRNA expression of apoptosis-related genes was measured by RT-qPCR and the results are shown in (**A**–**C**). (**D**) The mRNA expression of Rev-erbα during palmitate-induced apoptosis. Data are presented as mean ± SEM of three independent experiments. * *p* < 0.05.

**Figure 5 ijms-20-04524-f005:**
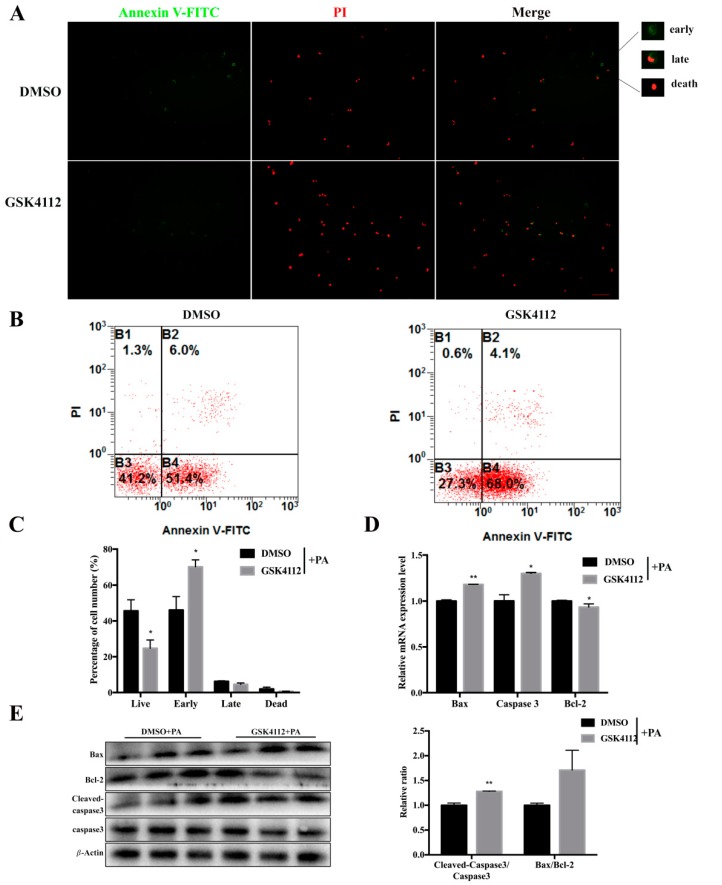
Rev-erbα agonist GSK4112 aggravated palmitate-induced preadipocyte apoptosis. 3T3-L1 cells were induced with 250 µM PA for 12 h, and then treated with 10 µM GSK4112 or DMSO for 24 h. (**A**) Annexin V-FITC/PI double staining was performed and detected by fluorescence microscopy, scale bar 100 µm. (**B**) Flow cytometry was used to analyze Annexin V-FITC/PI double staining of 3T3-L1 cells undergoing apoptosis. (**C**) The flow cytometry results are displayed as the percentage of the different stages of cell apoptosis. (**D**) RT-qPCR was used to detect mRNA levels of Bcl-2, Bax, and Caspase-3 after sequential treatment with PA and GSK4112. (**E**) (Left) Western blot analysis of apoptosis-related proteins after sequential treatment with PA and GSK4112. (Right) Results are represented as the mean values ± SEM. *n* = 3. * *p* < 0.05; ** *p* < 0.01.

**Table 1 ijms-20-04524-t001:** qRT-PCR primer sequences.

GeneName	Forward (5′-3′)	Reverse (5′-3′)
*Cyclin D*	TAGGCCCTCAGCCTCACTC	CCACCCCTGGGATAAAGCAC
*p27*	AGAAGCACTGCCGGGATATG	GACCCAATTAAAGGCACCGC
*Bmal1*	GGCTGTCATCATGAGCCTCT	TGAGGAAACACTGGAGCAGG
*Bcl-2*	GTCGCTACCGTCGTGACTTC	CAGACATGCACCTACCCAGC
*Bax*	TGAAGACAGGGGCCTTTTTG	AATTCGCCGGAGACACTCG
*Caspase-3*	ATGGAGAACAACAAAACCTCAGT	TTGCTCCCATGTATGGTCTTTAC
*Rev-erbα*	AACGGATGCTTGCCGAGAT	GGAGCCAGAGGTGGGATGT
β-*actin*	GTCCCTGACCCTCCCAAAAG	GCTGCCTCAACACCTCAACCC
β-*catenin*	TCCCATCCACGCAGTTTGAC	TCCTCATCGTTTAGCAGTTTTGT

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
