# Peer review of "Rev-erbα Inhibits Proliferation and Promotes Apoptosis of Preadipocytes through the Agonist GSK4112"

_ijms, 2019, doi:10.3390/ijms20184524_

Round 1
Reviewer 1 Report
The authors aimed to investigate whether extreme aging adipocytes undergo more apoptosis and produce more pro-inflammatory marker by regulating nuclear factor kappa- B (NF-κB) pathway. Aging adipocytes accumulated more lipids and produced more pro-inflammatory markers IL6 and TNF-α while underwent apoptosis during the 60 days. At the same time, aging adipocytes chronologically increased NF-κB p50 mRNA and protein expression while decreased IκBα protein level, indicating aged adipocytes produce more pro-inflammatory markers via regulating NF-κB pathway. In addition, similar results of pro-inflammatory markers and NF-κB pathway were observed in aging adipocytes in different microenvironments. Their results demonstrated that extremely aging adipocytes promote chronic inflammation via enhancing apoptosis and regulation of NF-κB signaling pathway. I am not happy with the use of 3t3 cell cultures. But the paper was done very well including the writing and the execution of all the procedures which were appropriate. This is a solid manuscript.
Minor points: line 61, continues ect: line 148, references needed here: line 167, with ?.
Author Response
Sorry, this comments are not for my study. For the language please see the attachment.

Reviewer 2 Report
Chu G and coworkers present the manuscript entitled “Rev-erbɑ inhibits proliferation and promotes apoptosis of preadipocytes through the agonist GSK4112”, in which it is described that GSK4112, a ligand of the nuclear receptor Rev-erbɑ, leading inhibition of Bmal1, resulting in the induction of apoptosis through augmenting Bax and Caspase 3 and decreasing Bcl2 expression, in a preadipocyte cell culture. The study design is accorded with the objectives. Furthermore, the conclusions are well supported by the experimental results. The study is novelty and provide interest data to the field. However some issues must be addressed:
Lines 65-71, at the end of the introduction: This part widely describes the results of the study, that is not the aim of the introduction section. The final paragraph is enough as conclusion of the study in the introduction. Figure 1B: change the vertical axis scale to better observe the significant differences: for example in the range 0.75-1.25. Figure 3B and 3D: Western blotting images are not significate of the histograms. It must be changed or repeated. The implication of the results from this study must be better correlated with the obese state in the discussion section.
Author Response
Lines 65-71, at the end of the introduction: This part widely describes the results of the study, that is not the aim of the introduction section. The final paragraph is enough as conclusion of the study in the introduction.
RESPONSE: We have deleted Lines 65-71.
Figure 1B: change the vertical axis scale to better observe the significant differences: for example in the range 0.75-1.25.
RESPONSE: We have changed the figure as the reviewer suggested.
Figure 3B and 3D: Western blotting images are not significate of the histograms. It must be changed or repeated.
RESPONSE: We have repeated the WB experiment and changed the figure.
The implication of the results from this study must be better correlated with the obese state in the discussion section.
RESPONSE: We have discussed the relationship between the results and the obese state.

Reviewer 3 Report
The manuscript by Chu et al. entitled “Rev-erbα Inhibits Proliferation and Promotes Apoptosis of Preadipocytes through the Agonist GSK4112” describes studies in 3T3-L1 cells examining the role of a nuclear receptor, known to be involved in the sleep-wake cycle, cell growth and cellular differentiation, in regards to cell proliferation and cell death. While the manuscript is interesting, the studies are quite limited scope as only a single agonist was tested against a single cell line. Additionally, no attempt to inhibit or knockdown the receptor as the activity of the Rev-erba is central to the authors conclusions. Overall, the authors should consider either knocking-down Rev-erba or using an antagonist to Rev-erbα to strengthen their conclusions. Below are specific concerns in regard to this study.
It is unclear why Bmal1 was chosen as the only marker for Rev-erba The inclusion of additional action markers, coupled with knocking-down and/or antagonist to Rev-erbα as mentioned above in the presence of GSK4112 will reinforce the authors conclusions. The authors use the term “vitality” (in text and Figure 1 legend) and “viability” (Figure 1 legend) in regards to the results shown with CCK-8. As these two words have distinctive meanings, the authors need to clarify what the conclusion is from the CCK-8 assay. Figure 2A is illegible as one cannot distinguish the EdU (red) or the Hoechst (blue). The same is true for Figure 5A illustrating the Annexin V assay. The cell cycle analysis (Figure 2) is not impressive in showing a clear inhibition of cell proliferation upon addition of the Rev-erbα agonist GSK4112. The authors should consider alternative methods to support their conclusion. It is unclear as the authors suggest that the palmitate-induced apoptosis (Figure 4) and the palmitate-induced apoptosis in the presence of the Rev-erbα agonist GSK4112 (Figure 5) actually results in the cells dying. The authors show changes in various apoptotic markers, however only externalize of the phosphotidylserine is observed in the absence of PI positive cells. A loss of membrane integrity at some level would be important to show to support the authors conclusions.
Round 2
Reviewer 3 Report
No comments to the authors.